# Prophage-triggered membrane vesicle formation through peptidoglycan damage in *Bacillus subtilis*

Masanori Toyofuku[1,2], Gerardo Cárcamo-Oyarce[2,5], Tatsuya Yamamoto[1], Fabian Eisenstein[3], Chien-Chi Hsiao[4], Masaharu Kurosawa[1], Karl Gademann [4], Martin Pilhofer[3], Nobuhiko Nomura[1] & Leo Eberl [2]

Bacteria release membrane vesicles (MVs) that play important roles in various biological processes. However, the mechanisms of MV formation in Gram-positive bacteria are unclear, as these cells possess a single cytoplasmic membrane that is surrounded by a thick cell wall. Here we use live cell imaging and electron cryo-tomography to describe a mechanism for MV formation in *Bacillus subtilis*. We show that the expression of a prophage-encoded endolysin in a sub-population of cells generates holes in the peptidoglycan cell wall. Through these openings, cytoplasmic membrane material protrudes into the extracellular space and is released as MVs. Due to the loss of membrane integrity, the induced cells eventually die. The vesicle-producing cells induce MV formation in neighboring cells by the enzymatic action of the released endolysin. Our results support the idea that endolysins may be important for MV formation in bacteria, and this mechanism may potentially be useful for the production of MVs for applications in biomedicine and nanotechnology.

[1] Department of Life and Environmental Sciences, University of Tsukuba, Tennoudai 1-1-1, Tsukuba, Ibaraki 305-8572, Japan. [2] Department of Plant and Microbial Biology, University of Zürich, Zollikerstrasse 107, Zürich 8008, Switzerland. [3] Department of Biology, Institute of Molecular Biology and Biophysics, ETH Zürich, Otto-Stern-Weg 5, Zürich 8093, Switzerland. [4] Department of Chemistry, University of Zürich, Winterthurerstrasse 190, Zürich 8057, Switzerland. [5] Present address: Department of Biological Engineering, Massachusetts Institute of Technology, Cambridge, MA 02139, USA. Nobuhiko Nomura and Leo Eberl jointly supervised this work. Correspondence and requests for materials should be addressed to M.T. (email: toyofuku.masanori.gf@u.tsukuba.ac.jp) or to M.P. (email: pilhofer@biol.ethz.ch) or to L.E. (email: leberl@botinst.uzh.ch)

Similar to eukaryotic cells, many bacteria release membrane vesicles (MVs) that transport cargos[1]. Bacterial MVs are involved in many fundamental biological processes, including horizontal gene transfer, virulence, phage decoy and cell-to-cell communication[2–5]. Bacterial MVs also play roles in carbon cycling in the marine ecosystem[6], affect human health due to their immunomodulatory activities, are used as vaccines and show potential as a platform for the development of cancer therapeutics and for applications in nanotechnology[7–9]. Despite their importance, little is known about the mechanisms of MV biogenesis. This is particularly true for MVs of Gram-positive bacteria[3], as the thick peptidoglycan (PG) cell wall, with a pore size of approximately 2 nm[10], is thought to prevent the release of MVs that typically have a diameter of 20–400 nm.

In contrast to Gram-positive bacteria, some progress has been made recently to understand MV biogenesis in Gram-negative bacteria. Several models for MV biogenesis have been proposed, suggesting that MVs are generated through different pathways, including localized membrane remodeling, altered turgor pressure of the periplasmic space, anionic charge repulsion between LPS molecules, depletion of the inner membrane and outer membrane linkage, and insertion of hydrophobic compounds in the outer leaflet of the membrane[4, 11–13]. MV production is stimulated by certain environmental conditions,

particularly by stress that causes DNA damage[14–16]. A recent study demonstrated that explosive cell lysis of a sub-population of cells leads to MV formation in *Pseudomonas aeruginosa*[17]. Explosive cell lysis is triggered by the induction of a phage-derived endolysin, which degrades the PG. Live cell imaging showed that cells with damaged PG explode and that the resulting shattered membranes fuse to form MVs. In biofilms and under stress conditions, including anaerobiosis and treatment with DNA-damaging agents, explosive cell lysis may play a major role in MV formation in *P. aeruginosa*[14, 17].

Here, we show that expression of an endolysin encoded by the defective prophage PBSX in the Gram-positive model organism *Bacillus subtilis* causes PG damage that leads to holes in the cell wall, through which vesicles are released. Our results, together with previous work[17], support a role for endolysins in MV formation in both Gram-positive and Gram-negative bacteria.

## Results

**MVs are released by dying cells.** The Gram-positive bacterium *B. subtilis* was previously demonstrated to produce MVs[18, 19]. To examine MV biogenesis, we followed the fate of single cells grown on solid medium pads. We observed that few cells released

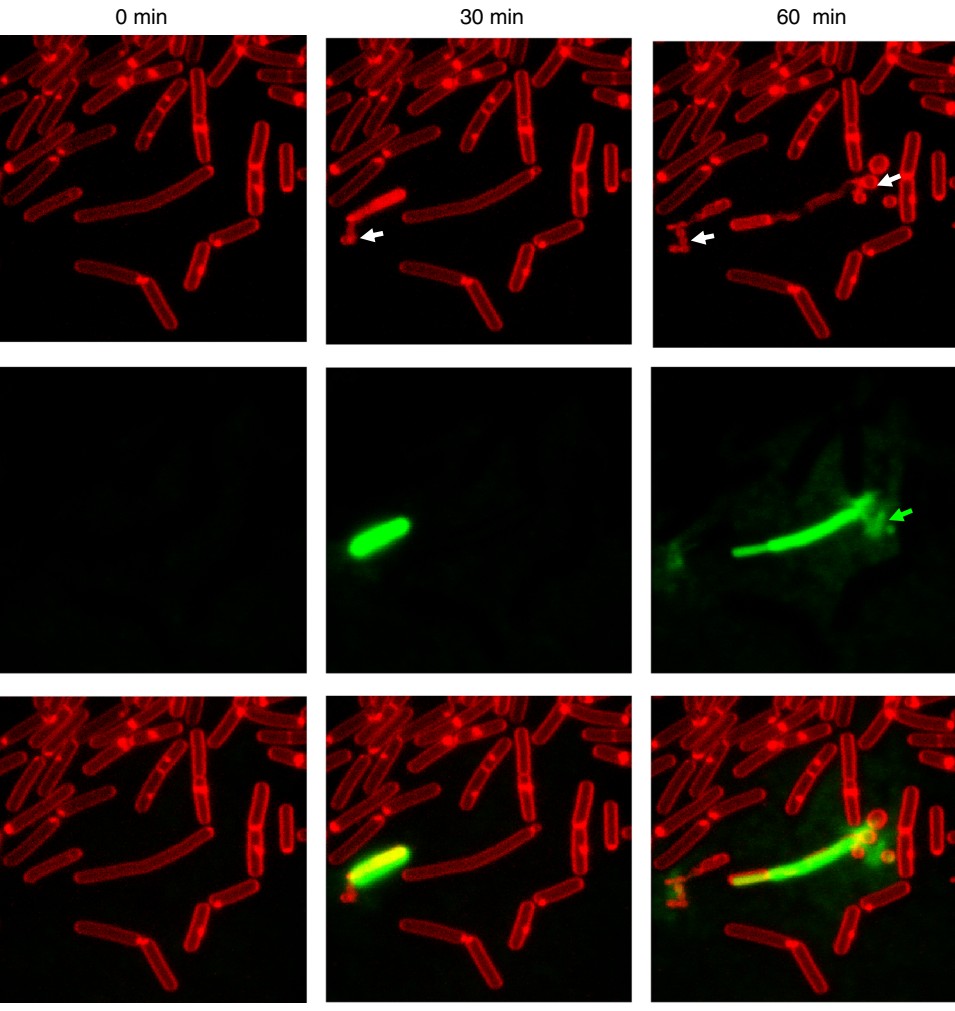

**Fig. 1** MVs are released by *B. subtilis* 168 at the onset of cell death. MV formation was followed at 30 min intervals at room temperature. Membranes were stained with the red fluorescent dye FM4-64 and SYTOX green was used to visualize dead cells and extracellular DNA (green). *White arrows* indicate cells releasing MVs and *green arrows* indicate extracellular DNA. *Scale bar*, 5 μm

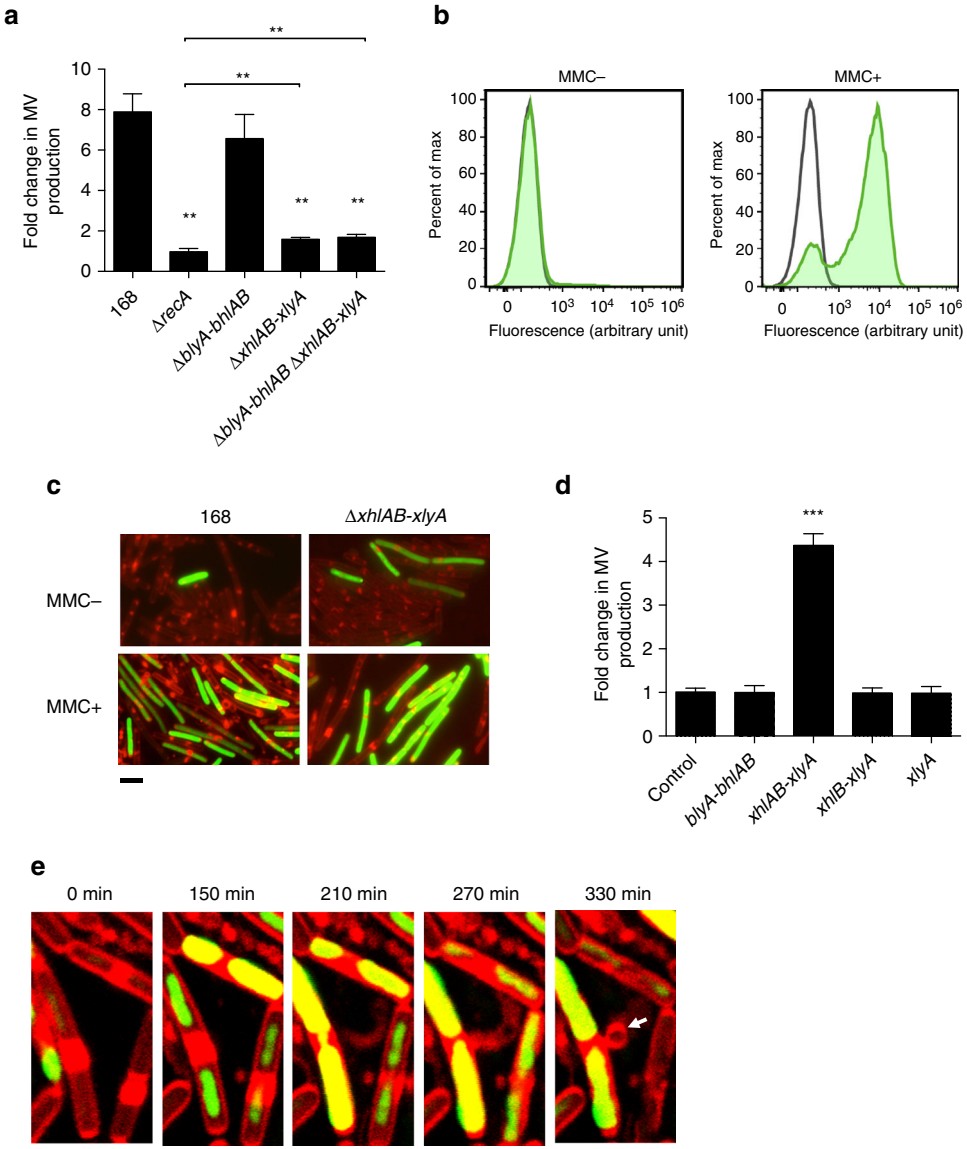

**Fig. 2** Genotoxic stress induces MV formation through activation of the holin–endolysin system. **a** Fold-change of MV production of MMC-treated cells relative to non-treated cells. $n = 3$; mean ± s.d. **$P < 0.01$ (unpaired $t$-test with Welch's correction). **b** Expression of the PBSX holin–endolysin gene cluster is induced by MMC. A transcriptional fusion of the $P_L$ promoter, which drives transcription of the PBSX late operon (including the holin–endolysin genes), to *zsGreen* was quantified by flow cytometry. *Green curve* shows activity of the $P_L$ promoter; *black curve* shows activity of the vector control. **c** Promoter ($P_L$) activities of the PBSX holin–endolysin gene cluster under MMC non-inducing and inducing conditions. The image shows FM4-64 (*red*) merged with ZsGreen (*green*). *Scale bar*, 5 μm. Phase contrast images along with the images of control experiments are shown in Supplementary Fig. 2a. **d** Fold-change of MV production by the expression of holin–endolysin genes relative to the control strain 168 ($P_{xylA}$). ***$P < 0.001$ (unpaired $t$-test with Welch's correction). **e** Live cell imaging of MV formation in *B. subtilis* 168 ($P_{xylA}$-*xhlAB-xlyA*) cells. *Arrows* indicate MVs. Cells were incubated on LB agarose pads containing 0.1% xylose, FM4-64 (*red*) and SYTOX green (*green*). *Scale bar*, 5 μm

large amounts of MVs. Those cells were stained by the membrane impermeable DNA dye SYTOX green, indicating that the cells' membranes were damaged and that DNA and cytoplasmic contents were released (Fig. 1). Twenty to 60% of the cells produced MVs within 12 h and all of these cells died ($n = 33$). These results indicate that the massive production of vesicles causes cell death.

**Genotoxic stress stimulates vesicle formation**. As only a subpopulation of the cells produced vesicles, we hypothesized that, similar to Gram-negative bacteria, genotoxic stress might induce MV formation[14, 17]. This was supported by the fact that MV formation was increased in the presence of the

DNA-damaging agent mitomycin C (MMC)[20] in a concentration-dependent manner (Fig. 2a and Supplementary Fig. 1a–c). Genotoxic stress response in bacteria involves the RecA pathway[21]. Hence, we tested a *B. subtilis recA* mutant for MV production. Production of MVs was impaired in the *recA* mutant (Fig. 2a) but could be rescued by genetic complementation (Supplementary Fig. 1d). These data suggest that RecA was only expressed in a subpopulation of the cells[22], and that MVs were primarily produced by this subpopulation (Fig. 1 and Supplementary Movie 1).

**Prophage PBSX is involved in MV release**. Previous work in *P. aeruginosa* has shown that a RecA-regulated endolysin, which

is part of a cryptic prophage region, can induce MV formation through enzymatic degradation of the PG layer[17]. The genome of *B. subtilis* 168 carries two prophages, SPβ and PBSX, and eight prophage-like regions (*pro*Φ1 to 7 and the skin element)[23]. PBSX is a defective prophage that consists of a head and a contractile phage tail. To examine whether a prophage region was involved in MV release, strains MS (missing the SPβ, PBSX and SKIN regions) and MGB469 (missing all prophage sequences except *pro*Φ7)[24] were tested for MV production. Both mutants produced reduced amounts of MVs (Supplementary Fig. 1e), indicating that genes required for MV formation were encoded within the SPβ or PBSX prophage regions. Importantly, the induction of the lytic cycle of these two prophages is controlled by RecA[20] and both loci encode a putative holin–endolysin system[25, 26].

Next, we constructed various SPβ and PBSX mutants to test the requirement of the different holin–endolysin systems. For the SPβ prophage, *blyA* (endolysin), *bhlA* (holin) and *bhlB* (holin) were deleted. For PBSX, *xhlA* (a membrane protein predicted to function synergistically with XhlB)[27], *xhlB* (holin) and *xlyA* (endolysin) were deleted. MV production was found to be decreased in the PBSX holin–endolysin mutant (Δ*xhlAB-xlyA*), while deletion of the SPβ holin–endolysin system (Δ*blyA-bhlAB*) did not affect MV production (Fig. 2a). Similar to *P. aeruginosa*[17], we did not see a difference in MV production between the wild type and the holin–endolysin mutant in the absence of MMC (Supplementary Fig. 1f). However, the importance of the holin–endolysin system was evident in the presence of MMC when MV formation was stimulated. The amount of MVs produced by the PBSX holin–endolysin mutant was higher than by a Δ*recA* mutant, suggesting that genes in addition to *xhlA*, *xhlB* and *xlyA* are involved in MV formation. Inactivation of the SPβ holin–endolysin system, in addition to the PBSX-encoded system (Δ*blyA-bhlAB* Δ*xhlAB-xlyA*), did not further reduce MV production (Fig. 2a). We used a transcriptional fusion of the PBSX holin–endolysin promoter (P$_L$ promoter)[28] to the reporter gene *zsGreen* to determine the fraction of cells that entered the lytic cycle. This analysis revealed that $5.22 \pm 0.50\%$ of the Δ*xhlAB-xlyA* mutant cells and $2.45 \pm 0.32\%$ of the wild-type cells were induced in liquid LB medium. The reduced number of fluorescent cells in the wild-type background is indicative of lysis due to the expression of the holin–endolysin system and the concomitant release of the fluorescent reporter protein ($P < 0.03$, unpaired *t*-test with Welch's correction) (Supplementary Fig. 2). The number of cells expressing the holin–endolysin genes increased upon exposure to MMC (Fig. 2b, c and Supplementary Fig. 2). MV-like structures were observed with $2.06 \pm 0.2\%$ of the wild-type cells, but only with $0.39 \pm 0.05\%$ of the Δ*xhlAB-xlyA* mutant strain when exposed to MMC for 1 h (Fig. 2c and Supplementary Fig. 2a).

**The PBSX endolysin triggers MV formation**. To control MV production in *B. subtilis* we constructed strains in which expression of *xhlA*, *xhlB*, or *xlyA* was under the control of a xylose-inducible promoter (Supplementary Fig. 3a). In agreement with a previous report on cell lysis in *B. subtilis*[27], we found that all three genes were required for MV production (Fig. 2d). In the mutant strain, in which all three genes were conditionally expressed, MV production and cell lysis was proportional to the amount of xylose added to the medium (Supplementary Fig. 3). Consistent with the mutant analysis, induction of the putative SPβ holin–endolysin genes (*blyA-bhlAB*) did not induce MV formation under the same conditions (Fig. 2d).

**MVs are released through holes in the cell wall**. Cell imaging revealed that induction of *xhlAB-xlyA* stimulated more cells to

release MVs when compared to the uninduced strain or the xylose-induced control strain (Fig. 2e, Supplementary Fig. 4 and Supplementary Movies 2 and 3). Most of the MV-producing cells were stained by SYTOX green or were ghost cells, indicating that the cells releasing MVs were dying (Supplementary Fig. 4a and Supplementary Movie 2). We observed that membrane material was extruded through the cell wall, rounded up and eventually detached from the cell wall (Fig. 2e and Supplementary Movies 2 and 3). Prolonged incubation with xylose resulted in the lysis of the majority of the cells, leaving membrane material, including MVs, behind (Supplementary Movie 2).

Initially, the release of MVs and DNA occurred from structurally intact cells (Figs 1 and 2e and Supplementary Movies 1 and 3), in contrast to the explosive cell lysis observed with *P. aeruginosa*[17]. Hence, while the holin–endolysin system induces cell lysis in both *P. aeruginosa* and *B. subtilis*, the underlying mechanisms of MV and DNA release are distinct, likely because of the differences in the cell envelopes. We reasoned that the thick PG layer of Gram-positive bacteria is initially only partially degraded and that MVs are released through gaps in the PG. To test this possibility, we expressed the *xhlAB-xlyA* genes in cells whose PG was labeled with fluorescein-D-lysine (FDL)[29]. Although induced cells released their cellular content, the PG of the ghost cells appeared to be largely intact (Fig. 3a and Supplementary Fig. 5). Using conventional transmission electron microscopy of thin sections, we observed holes in the cell wall (Fig. 3b). We also observed that the cellular content of these cells was less electron-dense compared to uninduced cells, indicating that they were ghost cells (Fig. 3b).

To image vesicle formation in a near-native state, we imaged frozen-hydrated cells by electron cryo-tomography (ECT) with a Volta phase plate (Figs 3c–k)[30]. Since ECT is limited by sample thickness and not applicable to wild-type *B. subtilis* cells, we generated a skinny Δ*ponA* mutant strain[31]. These cells had an average diameter of 700 nm and it was possible to clearly discern the cytoplasmic membrane surrounded by a ~55-nm-thick PG cell wall (Fig. 3c). A culture in which *xhlAB-xlyA* was induced for 3 h, 50% of the cells showed different stages of vesicle formation and/or cell lysis. Frequently, we observed one or several holes with 40–200 nm diameters in the cell wall. Such holes in the cell wall were generally accompanied by the expulsion of cytoplasmic membrane into the extracellular space (Figs 3d–h and Supplementary Movie 4). Similar observations have previously been reported for membrane protrusion in Gram-positive bacteria when their cell walls were treated with PG hydrolyzing enzymes[32, 33]. We also observed extracellular MVs which were not attached to any cell. Cells that were presumably in the early stage of vesicle formation exhibited a normal cytoplasmic appearance (Fig. 3f, g). By contrast, cells in presumably later stages showed significantly decreased cytoplasmic density and a massive formation of intracellular vesicles, while the cell wall remained mostly intact (Fig. 3d, h). ECT imaging of MMC-treated cells revealed that a subset of cells (~15%) showed similar signs of cell lysis and MV formation compared to the *xhlAB-xlyA* induced strain (Fig. 3i). In some of these cells we also detected phage-like structures with appearance and dimensions (sheath length: 210 nm, head diameter: 45 nm) similar to those reported for the PBSX phage[34] (Fig. 3j, k).

**MV formation triggers MV formation in neighboring cells**. During our live cell imaging experiments we noticed that cells producing MVs induced MV production in their neighboring cells (Figs 1, 4a and Supplementary Movies 1 and 5). We hypothesized that the endolysin that was released by

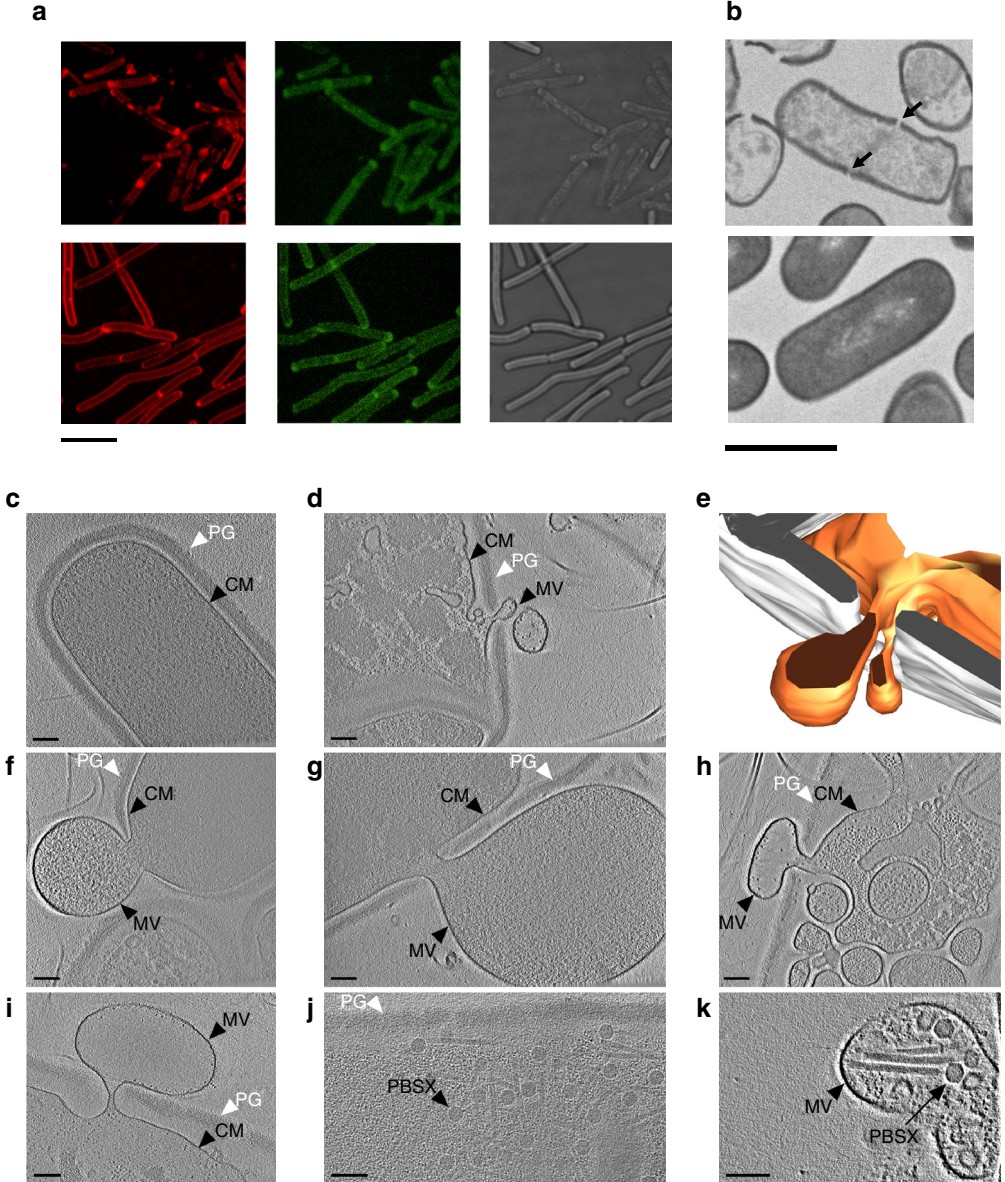

**Fig. 3** The activity of the PBSX endolysin creates holes in the bacterial cell wall. **a** Staining of the PG with FDL in cells expressing the holin–endolysin genes and in uninduced control cells. *Upper panel* shows strain 168 (P$_{xylA}$-*xhlAB-xlyA*), in which the holin–endolysin genes were expressed; the *lower panel* shows the control strain 168 (P$_{xylA}$). Membranes were stained with FM4-64 (*red, left*) and the PG was stained with FDL (*green, central*). Corresponding bright field images are shown in the *right panel*. *Scale bar*, 5 µm. **b** Thin section of induced *B. subtilis* 168 (P$_{xylA}$-*xhlAB-xlyA*) cells (*upper panel*) and of the control strain 168 (P$_{xylA}$) (*lower panel*). *Arrows* indicate holes in the bacterial cell wall. *Scale bar*, 1 µm. **c**–**k** ECT images of different *Bacillus ΔponA* strains. **c** Uninduced *B. subtilis ΔponA* (P$_{xylA}$-*xhlAB-xlyA*) cells have an intact peptidoglycan (PG) layer and cytoplasmic membrane (CM) and a dense cytoplasm. **d**–**h** Induced cells extrude MVs through holes in the PG layer. We also observed cells with lowered cytoplasmic densities that harbored intracellular vesicles (**h**). **i**–**k** Likewise, cells of a MMC-induced *B. subtilis ΔponA* culture showed MV formation (**i**) in addition to phage particles (PBSX) inside cells (**j**) and sometimes within extracellular MVs (**k**). Shown are 1.38 nm (**c**, **f**, **h**, **k**), 6.9 nm (**g**), 11.86 nm (**i**), 29.64 nm (**j**) thick slices through cryotomograms and a model (**e**) of the cell shown in **d**. *White*, peptidoglycan; *orange*, cytoplasmic membrane. *Scale bars*, 100 nm

dead cells may have weakened the PG of the neighboring cells and thus stimulated MV release. Incubation of paraformaldehyde-fixed *B. subtilis* cells with the supernatant of a PBSX endolysin-producing culture increased MV formation, while an uninduced culture supernatant or a heat-treated induced culture supernatant did not (Fig. 4b). This suggests that the PBSX endolysin can also trigger MV formation by degrading the PG of Gram-positive cells. In support of this hypothesis and in agreement with previous work[32, 33, 35] we observed that cell wall-weakening treatment with lysozyme stimulated MV formation (Fig. 4c). MVs purified from an endolysin-producing culture also induced MV formation, indicating that at least part of the released endolysin is associated with MVs (Fig. 4d).

## Discussion

Given that the cell walls of Gram-negative and Gram-positive bacteria are fundamentally different, it is believed that different mechanisms must account for the biogenesis of MVs[2, 3]. While several mechanisms for vesicle biogenesis have been demonstrated for Gram-negative bacteria[2, 11, 17] little is known about Gram-positive bacteria[3]. Here we showed that similar to

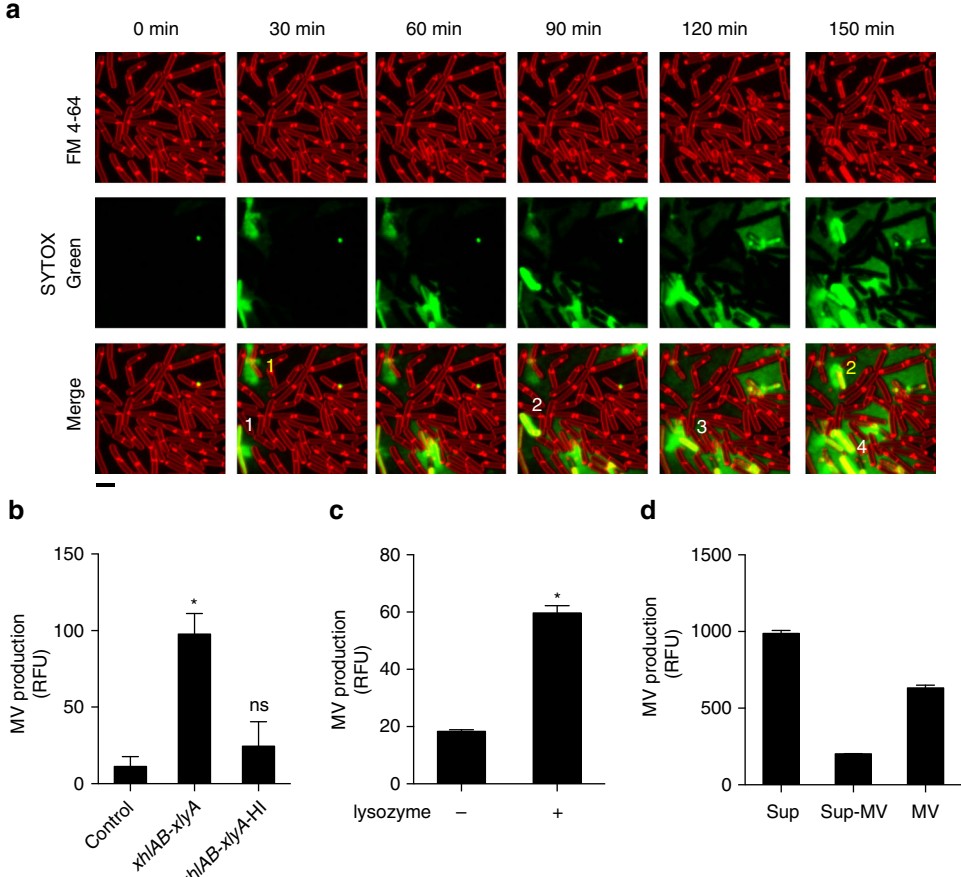

**Fig. 4** Endolysin triggers MV release in neighboring cells. **a** Cell death induces MV release in neighboring cells. Numbers depict the progression of cell death in the population. *Scale bar*, 5 μm. Membranes are stained with the red fluorescence dye FM4-64. Dead cells and extracellular DNA are visualized with SYTOX green. **b** MV production is increased in the presence of PBSX endolysin. Cells were stained with FM4-64FX, fixed and incubated with supernatants of the control strain 168 (P$_{xylA}$) or with induced 168 (P$_{xylA}$-xhlAB-xlyA) supernatants (untreated; xhlAB-xlyA and heat-inactivated; xhlAB-xlyA-HI). n = 6; mean ± s.d. ****P < 0.0001. **c** MV production in strain 168 is stimulated by treatment of exponentially grown cells with lysozyme. n = 3; mean ± s.d. ****P < 0.0001, or **d** in the presence of purified MVs. Cells were stained with FM1-43FX, fixed and incubated with the supernatant (Sup), the supernatant without MVs (Sup-MV) and purified MVs of induced 168 (P$_{xylA}$-xhlAB-xlyA) cultures. n = 3

the Gram-negative bacterium *P. aeruginosa*[17], MV formation under DNA stress in the Gram-positive bacterium *B. subtilis* is stimulated by the expression of an endolysin encoded by a defective prophage. Although in both organisms the enzymatic activities of the endolysins weaken the PG, the consequences are different: while *P. aeruginosa* cells round up and explode with MVs being formed from scattered membrane fragments, *B. subtilis* cells protrude cytoplasmic membrane material through holes in the PG and these membrane blebs are then released as MVs. While in this process *P. aeruginosa* cells completely disintegrate (explosive cell lysis), the *B. subtilis* cell wall does not entirely disintegrate although most cells die due to the loss of cytoplasmic membrane integrity as indicated by the formation of ghost cells and intracellular MVs (Fig. 5).

Although our study shows that MV biogenesis can involve cell lysis in Gram-positive bacteria, it is likely that other mechanisms exist that are not dependent on cell death[3, 36]. However, it is interesting to note that a recent paper demonstrated that vesicles produced by *Streptococcus pneumoniae* were specifically enriched for a putative phage-associated endolysin[37]. More important in the context of our study is the finding that a proteome analysis of vesicles produced by *B. subtilis* in standard laboratory conditions identified various phage proteins[19]. This enrichment was particularly high for sporulating *B. subtilis* cultures. It is also interesting to note that bacterial phage endolysins have been used

as antibiotics and it has been observed that endolysin treatment leads to membrane extrusions through holes[32, 33, 38], which is in agreement with our study. Yet, these findings were not linked to MV formation at that time.

Given that bacteriophages are the most abundant organisms in the biosphere and they are ubiquitous[39], we speculate that phage-triggered cell death may be an underestimated mechanism for vesicle biogenesis in nature. This hypothesis is supported by previous studies that have demonstrated that vesicles can harbor complete viral genomes[40] and that DNA associated with MVs isolated from open ocean samples are highly enriched for viral sequences[6, 41]. Our finding of phages inside extracellular vesicles (Fig. 3k) might indicate that phages themselves are released inside MVs. Moreover, UV radiation of water samples, a trigger for prophage induction, was demonstrated to greatly stimulate vesicle formation[42].

Although endolysin-triggered vesicle formation is linked to the death of some cells, this may provide a benefit to the rest of the population. This concept is similar to pyocins in *P. aeruginosa*, which require cell lysis to be released but protect the remaining population from susceptible bacterial competitors[43]. Likewise, cell lysis is required for DNA release, but this in turn promotes the formation of biofilms, where the remaining bacteria show greatly increased resistance to various stresses[17]. Previous work has demonstrated that MV production is an important factor in

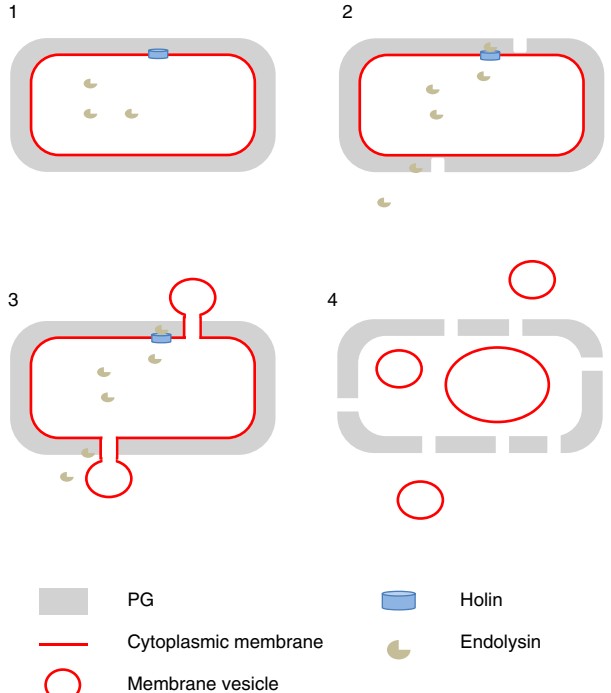

**Fig. 5** Schematic model of MV production by the holin–endolysin pathway in *B. subtilis*. Step 1, the holin forms pores in the cytoplasmic membrane to allow the endolysin to access the PG. Step 2, the endolysin degrades the PG and creates holes. The PG can also be degraded from outside, e.g., by endolysins released from neighboring cells. Step 3, the cytoplasmic membrane is extruded through the holes in the PG, presumably due to high cell turgor. Step 4, the MVs are released. We also observed that intracellular membrane fragments can vesicularize and may eventually be released from the dying cell

neutralizing environmental agents that target the outer membrane of Gram-negative bacteria, such as antimicrobial peptides or bacteriophages[44]. Another intriguing interplay between lytic phages and vesicles has recently been discovered for *B. subtilis* and its phage SPP1[45]. In this study, it was demonstrated that vesicles that are released from phage-sensitive cells harbor phage receptors, which upon fusion with resistant cells render them sensitive to the phage. This phenomenon can occur in an interspecies manner, thus facilitating the attachment of phages to non-host species and thereby promoting phage spread and invasion in mixed natural communities. As this facilitates transduction and consequently horizontal gene transfer among different bacterial species, this phenomenon may have a major impact on bacterial evolution.

We further propose that PG hydrolyzing enzymes may play a major role in MV formation in bacteria. Such enzymes are widespread in nature and are not only associated with lytic phages but also with bacteriocin biosynthesis clusters and effectors of type 6 secretion systems[46, 47]. Future studies will elucidate which proportion of the MVs in the environment originate from cell lysis or are produced by other mechanisms and whether they differ in their composition. Engineered strains, in which expression of an endolysin can be conditionally induced, may allow for the mass production of MVs for applications in biomedicine and nanotechnology.

## Methods
**Growth conditions**. *B. subtilis* was routinely cultured in Luria-Bertani (LB) broth. Antibiotics were used as required at the following concentrations: kanamycin 5 µg/mL, chloramphenicol 5 µg/mL, erythromycin 0.5 µg/mL and spectinomycin 100 µg/

mL. For MV assays, *B. subtilis* was inoculated at an optical density of 0.01 at 600 nm ($OD_{600}$) in LB medium. MMC or xylose was added to the medium at the exponential phase ($OD_{600} = 0.3$–0.4). MMC was added at 12.5 ng/mL and xylose was added at 0.013%, unless stated. Fluorescent stains used in this study were the lipophilic membrane stains FM4-64, FM4-64FX and FM1-43FX (1 µg/mL), the extracellular DNA and dead cell stain SYTOXgreen (5 µM), the DNA stain DAPI (1 µg/mL). PG staining was done with the aid of FDL (10 µg/mL), which was synthesized according to Kuru et al.[29, 48] FDL was added prior to inoculation of the culture, or 30 min prior to adding MMC or xylose. Cells were harvested and washed twice in phosphate-buffered saline (PBS) to remove excess FDL before analyses. Lysozyme from chicken egg white (1,000,000 units/mg; SERVA, Germany) was used for lysozyme treatment of the cells. Exponentially grown cells (OD600 = 0.3–0.4) were treated with 1.25 µg/mL of lysozyme for 3 h.

**Bacterial strains and plasmids**. Strains used in this study are listed in Supplementary Table 1. Primers used in this study are listed in Supplementary Table 2.

The SPβ holin–endolysin deletion strain 168 Δ*blyA-bhlAB* was constructed as follows. The *blyA* upstream and the *bhlB* downstream regions were amplified by PCR using the TYP58/TYP62 and TYP63/TYP59 primer sets, respectively. An erythromycin-resistance gene was amplified using TYP61 and TYP 62 primers and the pMutinNC plasmid[49] as a template. The three PCR products were ligated by overlap PCR using primers TYP57 and TYP59. The ligated PCR product was transformed into *B. subtilis* 168 by the natural competence method[50], and transformants were selected with 0.5 µg/mL erythromycin. The *xhlAB-xylA* deletion strain Δ*xhlAB-xlyA* was constructed in the same manner using the following primers: TYP65 to TYP72. *B. subtilis* TMO310[51] was used as a template for the spectinomycin-resistance gene. The *blyA-bhlAB* and *xhlAB-xlyA* deletion strain Δ*blyA-bhlAB*Δ*xhlAB-xlyA* was constructed by transforming strain 168Δ*xhlAB-xlyA* with chromosomal DNA extracted from strain 168Δ*blyA-bhlAB*.

The strain with the xylose-inducible promoter cassette, strain 168 ($P_{xylA}$), was constructed as follows. The *amyE* 5′-*rrnB* terminator region was amplified by PCR using TYP2 and TYP82 primers from chromosomal DNA extracted from *B. subtilis* TAY3000[52]. The *xylR*-$P_{xylA}$, *amyE* 3′ and *spoVG* terminator regions were PCR amplified with the primer sets TYP81/TYP84, TYP105/TYP4 and TYP83/TYP20, respectively. The chloramphenicol-resistance gene was amplified using primers TYP106 and TYP107 from plasmid pDH88[53]. The PCR products of the *spoVG* terminator and the chloramphenicol-resistance gene were digested by *Eco*RI, and ligated using the DNA Ligation Kit Mighty Mix (Takara Bio, Japan). The ligated product was amplified by PCR using primers TYP83 and TYP4. The *amyE* 5′-*rrnB* terminator, *xylR*-P*xylA*, the *spoVG* terminator-chloramphenicol-resistance gene and the *amyE* 3′ regions were ligated by overlap PCR with primers TYP1 and TYP3. The PCR product was introduced into the *amyE* locus of the *B. subtilis* 168 by double-crossover recombination to generate strain 168 ($P_{xylA}$). Strain 168 ($P_{xylA}$-*blyA-bhlAB*) was constructed as follows. The *amyE* 5′-*rrnB* terminator-*xylR*-P$_{xylA}$ and *spoVG* terminator-chloramphenicol-resistance gene-*amyE*3′ regions were PCR amplified with primer sets TYP2/TYP86 and TYP87/TYP4. The SPβ holin and endolysin genes, *blyA*, *bhlA* and *bhlB*, were amplified with primers TYP85 and TYP88. These three PCR products were ligated by overlap PCR with primers TYP1 and TYP3, and the PCR product was introduce into the *amyE* locus of the *B. subtilis* 168. The strains 168 ($P_{xylA}$-*xhlAB-xlyA*), 168 ($P_{xylA}$-*xhlB-xlyA*) and 168 ($P_{xylA}$-*xlyA*) were constructed in the same manner with following primers: TYP89 to TYP92 for strain 168 ($P_{xylA}$-*xhlAB-xlyA*), TYP91 to TYP94 for strain 168 ($P_{xylA}$-*xhlB-xlyA*), TYP91, TYP92, TYP101 and TYP102 for strain 168 ($P_{xylA}$-*xlyA*), TYP1 to TYP4 for strain 168 ($P_{xylA}$-*xhlAB-xlyA*), strain 168 ($P_{xylA}$-*xhlB-xlyA*) and strain ($P_{xylA}$-*xlyA*).

The *recA* deletion strain Δ*recA* was constructed as follows. The *recA* upstream and downstream regions were PCR amplified with primer pairs TYP109/TYP113 and TYP114/TYP111, respectively. The chloramphenicol-resistance gene was amplified using primers TYP112 and TYP115 and plasmid pDH88 as a template. These three PCR products were ligated by overlap PCR using primers TYP108 and TYP110, and the ligated product was transformed into *B. subtilis* 168 to generate the mutant strain 168 Δ*recA*. The *recA* gene was amplified with its native promoter and terminator by PCR using TYP116 and TYP117 primers, and the PCR product was digested with *Bam*HI and *Eco*RI. The *amyE* 5′-*rrnB* terminator and the DNA region containing the kanamycin-resistance and *amyE*3′ genes were amplified by PCR using the primer sets TYP2/TYP7 and TYP8/TYP3, respectively. The amplified *amyE* 5′-*rrnB* terminator and the kanamycin-resistance gene-*amyE*3′ regions were digested with *Bam*HI and *Eco*RI, respectively. These three DNA fragments were ligated using the DNA Ligation Kit Mighty Mix. The ligated product was PCR amplified with primers TYP1 and TYP3. The PCR product was introduced into the *amyE* locus of the *B. subtilis* 168 strain to generate *amyE*::*recA* for *recA* complementation. The *amyE* 5′-*rrnB* terminator-kanamycin-resistance gene-*amyE*3′ region was PCR amplified using primers TYP1 and TYP3 and TAY3000 as a template, and the PCR product was transformed into *B. subtilis* 168 to generate strain *amyE*::*kan*. The *recA* gene of *amyE*::*recA* and *amyE*::*kan* was deleted by transformation with chromosomal DNA extracted from the Δ*recA* mutant strain.

The pHY300-P*veg*-*egfp*-term plasmid was constructed as follows. The *rpsD* terminator, *veg* promoter and *B. subtilis spoVG* terminator were amplified by PCR

### Legend
PG
Holin
Cytoplasmic membrane
Endolysin
Membrane vesicle

from *B. subtilis* chromosomal DNA using the primer sets TYP118/TYP119, TYP120/TYP121 and TYP123/TYP124, respectively. The *egfp* was PCR amplified from plasmid pEGFP with primers TYP125 and TYP122. These four PCR products were ligated by overlap PCR with TYP118 and TYP124 primers. The ligated product was digested with *Bam*HI and *Eco*RI and ligated into pHY300PLK cut with the same enzymes, using the DNA Ligation Kit Mighty Mix. The plasmids pHY300-P$_L$-egfp-term and pHY300-P$_L$-egfp-term were constructed in the same manner with the following primers: TYP118, TYP119 and TYP122 to TYP127 for pHY300-P$_L$-egfp-term, TYP118, TYP119, TYP122, TYP123 and TYP128 for pHY300-egfp-term. P$_L$ is the promoter region of the PBSX late operon that contains the holin–endolysin genes[27, 28]. Plasmid pHY300-P$_L$-ZsGreen-term was constructed as follows. The *zsGreen* gene was amplified by PCR from plasmid pZsGreen using primers TYP129 and TYP130. The *spoVG* terminator and P$_L$-rpsD terminator regions were PCR amplified from plasmid pHY300-P$_L$-egfp-term with primer pairs TYP131/TYP133 and TYP132/TYP134, respectively. These three PCR products were ligated by overlap PCR using primers TYP118 and TYP119. The ligated product was digested with *Bam*HI and *Eco*RI and ligated into pHY300PLK cut with the same enzymes. Plasmid pHY300-P*veg*-ZsGreen-term was constructed as follows. The *spoVG* terminator-ZsGreen region was PCR amplified from plasmid pHY300-P$_L$-ZsGreen-term using primers TYP133 and TYP136. The P*veg* promoter-*rpsD* terminator region was amplified from plasmid pHY300-P*veg*-egfp-term with primers TYP134 and TYP135. These two PCR products were ligated by overlap PCR using primers TYP118 and TYP124. The ligated product was digested with *Bam*HI and *Eco*RI and ligated into pHY300PLK cut with the same enzymes. The pHY300-ZsGreen-term was constructed in the same manner using primers TYP118, TYP124, TYP133, TYP134, TYP136 and TYP137. The plasmids were transformed into *B. subtilis* 168 and the Δ*xhlAB-xlyA* mutant.

The *ponA* deletion mutants were constructed as follows. The *ponA* upstream and downstream regions were amplified by PCR with TYP139/TYP144 and TYP141/TYP145 primer sets, respectively. The kanamycin-resistance gene was amplified by using TYP143 and TYP146 primers from TAY3000. These three PCR products were ligated by overlap PCR with TYP140 and TAY142 primers, and the ligated product was inserted into *B. subtilis* 168 by transformation to generate Δ*ponA*. The *ponA* of 168 (P$_{xylA}$-xhlAB-xlyA) were deleted by transforming with chromosomal DNA extracted from Δ*ponA*. Transformation was confirmed by PCR and DNA sequencing.

**Flow cytometry.** *B. subtilis* was grown in test tubes containing 4 mL of LB medium supplemented with 10 μg/mL tetracycline at 37 °C, under shaking conditions. Upon reaching an OD$_{660}$ of 0.3, MMC was added to a final concentration of 100 ng/mL. Following 1.5 h of incubation, cells were fixed with 3.7% formaldehyde for 15 min. The fixed cells were harvested by centrifugation, washed, resuspended in PBS and used for flow cytometry (LE-SH800ZFP, SONY). For each sample, 100,000 events were recorded for analysis. The data were analyzed using the FlowJo software (FlowJo, LLC).

**MV isolation and quantification.** MVs were isolated and quantified as described previously[17]. Briefly, culture supernatants were filtered with a 0.4-μm pore size PVDF filter (Merck Millipore, Germany), and ultracentrifuged for 1 h at 150,000 × *g*, 4 °C. The pellet was resuspended in double distilled water for MV quantification. For further purification, MV pellets were resuspended in 45% ioxidanol (Optiprep, AXIS-SHIELD, Scotland) in HEPES-NaCl buffer and subjected to density gradient ultracentrifugation in an iodoxinal gradient of 45–10%. For MV quantification, MVs were stained with the fluorescent dye FM4-64 or FM1-43 (Life Technologies, USA)[17]. Fluorescence was quantified using a Varioscan flash fluorimeter (Thermo Scientific, USA) or a Synergy HT plate reader (MWG Biotech, Germany).

**MV induction using the supernatant of *xhlAB-xlyA* induced cells.** The *xhlAB-xlyA* inducible strain, 168 (P$_{xylA}$-xhlAB-xlyA) or the control strain, 168 (P$_{xylA}$), was incubated to OD$_{600}$ of 0.3–0.4 and 0.1% xylose was added. The strains were further incubated for 4 h and then the supernatant was collected. The supernatants were filtered with a PES 0.22 μm filter (Merck Millipore, Ireland). FM4-64FX or FM1-43FX dye fixed late exponential phase *B. subtilis* 168 was added to the supernatants to examine if MVs can be induced from these cells by exposure to the supernatants. Cells stained with FM4-64FX or FM1-43FX were fixed with 4% paraformaldehyde and the cells were washed twice with PBS after fixation. Heat treatment of the *xhlAB-xlyA* induced supernatant was carried out at 95 °C for 10 min. The FM dye fixed cells were incubated with the supernatants or MVs at 37 °C for 16 h. After 16 h incubation, the cells were pelleted down and the MVs in the supernatant were isolated by ultracentrifugation as described above. MVs derived from the FM dye fixed cells were quantified by detecting the FM dyes as described for the MV quantification.

**Microscopy.** Confocal laser scanning microscopy (CLSM) was performed with a DM5500Q microscope with Leica Application suite (Leica, Germany)[54] or with a LSM710 microscope operated with Axiovision System (Carl Zeiss, Germany)[14]. Pictures were processed using Fiji[55] or IMARIS (Bitplane, Switzerland). For CLSM, cells were grown on 0.5% agarose pads supplemented with LB. Pads were mounted

in press-to-seal silicon isolators (Sigma-Aldrich, USA) and supplemented with fluorescent dyes (1 μg/mL for FM4-64, 5 μM for SYTOX green and 1 μg/mL for DAPI), MMC (5 ng/mL) or xylose (0.1% w/v) when required. For live cell imaging of cells under non-treated conditions, *B. subtilis* 168 cells were stained with FM4-64 before they were transferred to LB agarose pad containing SYTOX green and FM4-64. The cells were observed under room temperature following incubation for 5 h at 37 °C. For live cell imaging of *xhlAB-xlyA* inducible strains, xylose was added to the medium at the exponential phase (OD$_{600}$ = 0.3–0.4) and were incubated for 2 h before they were stained with FM4-64 and inoculated on LB agarose pads containing 0.1% xylose, FM4-64 and SYTOX green. The cells were observed under room temperature.

For PG staining, cells were grown in the presence of FDL (10 μg/mL). After incubation, the cell culture was centrifuged and the supernatant was removed carefully. The cells were resuspended in LB before mounting on LB agarose pads for microscopic inspection.

For scanning electron micropy, *B. subtilis* 168 (P$_{xylA}$) and 168 (P$_{xylA}$-xhlAB-xlyA) were grown in test tubes containing 4 mL of LB medium at 37 °C to an OD$_{660}$ of 0.3, and then xylose was added to a final concentration of 0.1% (w/v). Following a 1 h incubation, the cells from 1 mL culture were harvested and fixed in 2% glutaraldehyde, 0.1 M phosphate buffer (pH7.0) at 4 °C overnight. The fixed cells were attached to poly-L-lysine-coated coverslips and dehydrated sequentially with 50, 70, 90 and 99.5% ethanol. The coverslips were mounted onto aluminum stabs, dried using a HCP-2 critical-point dryer (Hitachi Ltd, Japan) and sputter coated with platinum using an E-1030 ion sputtering machine (Hitachi Ltd). The cells were inspected using a HITACH-S-4200 scanning electron microscope (Hitachi Ltd).

For transmission electron microscopy (TEM), purified MVs were stained with uranyl acetate and inspected using a JOEL JEM 2000EX transmission electron microscope (Hanaichi Ultrastructure Research Institute, Japan). For TEM ultra thin section, *B. subtilis* was fixed with 2% glutaraldehyde/100 mM phosphate buffer followed by 2% osmium tetroxide/100 mM phosphate buffer, washed with 100 mM phosphate buffer and dehydrated sequentially with 50, 70, 90 and 100% ethanol. The sample was soaked in propylene oxide and mixed with propylene oxide:resin (7:3) for infiltration. Epon 812 was used for embedding the sample. Uranyl acetate and lead was used for staining.

**Electron cryo-tomography.** For ECT imaging of vesicle formation, a Δ*ponA* mutant of the *xhlAB-xlyA* inducible *B. subtilis* 168 P$_{xylA}$-xhlAB-xlyA strain was cultured to an OD$_{600}$ of 0.3–0.4. Cells were induced for 3 h with xylose (0.013% w/v) before plunge freezing. For ECT imaging of induced PBSX phages, *B. subtilis* 168 Δ*ponA* was cultured for 2 h before adding 25–50 ng/mL MMC. Cells were incubated for another 4 h before plunge freezing. Plunge freezing and ECT imaging were performed according to Weiss et al.[56] Cell suspensions were mixed with bovine serum albumin-coated 10-nm colloidal gold, and applied to glow-discharged EM grids (R2/1 copper, Quantifoil). Samples were vitrified in liquid ethane–propane using a Vitrobot Mark IV (FEI). ECT data were collected on a Titan Krios (FEI) or a Polara (FEI) transmission electron microscope equipped with imaging filter (Gatan, slit with 20 eV) and K2 Summit direct electron detector (Gatan). Tilt series were acquired with the software SerialEM[57] or UCSF Tomo[58]. The angular range was −60° to +60° and the angular increment was 1°. The total electron dose was 90–100 electrons per Å². Images were recorded in focus with a Volta phaseplate (FEI) or without phaseplate at 10 μm under-focus. Tilt series were aligned using gold fiducials or patch tracking. Three-dimensional reconstructions were calculated by weighted back projection using IMOD[59]. Segmentations and visualization were performed using IMOD.

**Procedure for the preparation of FDL.** The compound was prepared by a published procedure, which was slightly modified (Supplementary Fig. 6)[48]. Exposure of the reaction to light was avoided throughout the procedure. A Schlenk flask was charged with N-Boc-D-Lys-OH (0.277 mmol) (Sigma-Aldrich, Switzerland) and fluorescein isothiocyanate (0.231 mmol) (Sigma-Aldrich, Switzerland) in dry dimethylformamide (DMF) (4.0 mL), and the reaction mixture was stirred for 4 h at room temperature in a nitrogen atmosphere. The solvent was removed directly under high vacuum (ca 0.1 mbar). The residue was dissolved in EtOAc (25 mL) and washed with 1 N HCl (10 mL) and brine (10 mL) solution, dried over Na$_2$SO$_4$ and concentrated *in vacuo*. The crude intermediate was stirred in trifluoroacetic acid (TFA)/water (10 mL, 1:1) for 1 h and the solvent was removed under reduced pressure. The residue was purified by reverse phase-HPLC and the pure fractions were lyophilized to afford 67 mg of the desired compound as a light orange solid (56% yield). For reverse phase-HPLC, 15–95% CH$_3$CN/H$_2$O was used for elution with a flow rate of 1.0 mL/min.
The $^1$H-NMR spectrum of FDL is in agreement with values reported in the literature[29].
$^1$H NMR (400 MHz, CD$_3$OD): δ 8.12 (d, *J* = 1.9 Hz, 1H), 7.74 (d, *J* = 8.2 Hz, 1H), 7.16 (d, *J* = 8.2 Hz, 1H), 6.71–6.63 (m, 4H), 6.54 (dd, *J* = 8.8, 2.4 Hz, 2H), 3.65 (br s, 2H), 3.59 (t, *J* = 6.1 Hz, 1H), 2.01–1.84 (m, 2H), 1.79–1.66 (m, 2H), 1.60–1.46 (m, 2H).

**Data availability**. The authors declare that all the relevant data supporting the findings of the study are available in this article and its Supplementary Information files, or from the corresponding authors upon request.

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

## Acknowledgements

M.T. was supported by a Grant-in-Aid for Scientific Research (25701012, 16K14795 and 16H06189) from the Ministry of Education, Culture, Sports, Science and Technology of Japan (MEXT), and from Japanese Society for the Promotion of Science (JSPS) Post-doctoral Fellowship for Research Abroad. N.N. was supported by the Japan Science and Technology Agency, ERATO (JPMJER1502), and a Grant-in-Aid for Scientific Research (60292520) from MEXT. L.E. was supported by the Swiss National Science Foundation (SNSF) (Project 3100A0-104215) and C.-C.H. by SNSF (Sinergia CRSII3_154430). M.P. was supported by SNSF (grants 31003A_152878, 316030_164092) and the European Research Council (STG 679209). ECT data were recorded at the ETH platform ScopeM and at the University of Zürich. We thank Kirsty Agnoli-Antkowiak for valuable comments on the manuscript. We acknowledge the National BioResource Project (NBRP) Japan for kindly providing MS and MGB469 strains.

## Author contributions

M.T., G.C.-O., T.Y., F.E., K.G., M.P. and L.E. designed the experiments; M.T., G.C.-O., T.Y., F.E., C.-C.H. and M.K. conducted experiments; all authors analyzed the results; M. T. and L.E. wrote the manuscript with support from G.C.-O., T.Y., F.E., C.-C.H., K.G., M. P. and N.N.; all authors reviewed the manuscript.

## Additional information

**Competing interests:** The authors declare no competing financial interests.

