## [Peer Review File · Nature Communications]

Reviewers' comments:

Reviewer #1 (Remarks to the Author):

The paper by Toyofuku et al describes phage mediated lysis in *B. subtilis* and goes on to claim that this is a mechanism for extracellular vesicle formation. The paper does a disservice to the field by mixing up very different processes and bringing them together in a form that will sow confusion. Phage mediated lysis will result in damaged membranes, cell lysis and release of membrane fragments that can reassemble into structures that resemble vesicles as a function of the tendency of lipids to form bilayers and vesicles (like is seen in liposomes). However, to claim that this lysis is a mechanism for the phenomenon of extracellular MV vesicle formation that has now been described for many gram positive organisms is to jump to a conclusion that is not supported by the data. Hence, this paper should not be published in its current form. In essence there are serious concerns about the data shown in this paper and the extrapolations and conclusions made from those results to MV biogenesis are not supported by the data.

Specific comments

1. The title is misleading for it implies that MVs come from cell lysis. Many studies of gram positives have shown that MVs from gram positives are made by an active process from live cells. Phage-triggered lysis will produce vesicles from membrane association but the authors have produced no biochemical evidence that these vesicles are the same that are produced by non-lytic cells.
2. Line 180 states 'In support of this hypothesis and in agreement with previous work^{23,24,26} we observed that cell wall weakening treatment with lysozyme stimulated MV formation' – this is circular reasoning: lysozyme is known to lyse cells releasing membranes that can assemble into structures resembling vesicles. The same can be achieved by sonicating cells.
3. Figure 1 purports to show vesicles. What is the evidence that the structures pointed to by the arrows are vesicles? I can easily argue that these are sectional fractions of cells that appear smaller by microscopy. Figure 1 data is not convincing. This reviewer has the same concern for Figure 2e – I can argue that what appears to be vesicle is nothing more than a cross-section of a bacterium that was in the background and has moved into view. The microscopy shown is substandard for the claims being made.
4. Extended figure 3d purports to show MVs by EM. These images do not look like the spherical vesicle populations described in other papers of gram+ MV. To this author these structures look like lysed membrane fragments, which are very different than MVs.
5. The term 'bubbling cell death' makes no sense. What the authors are describing is bacterial cell lysis from phage and/or endogenous lysins...there is no need for additional confounding terminology.

Reviewer #2 (Remarks to the Author):

The authors describe an interesting mechanism for MV formation in *B. subtilis*. They show the formation of so called pinholes in the PGN of prophage PBSX containing cells of *B. subtilis*. The prophage encoded endolysin gene is SOS-inducible and responsible for pinhole formation that causes finally cell death due to cytoplasmic membrane damage. The formed MV can induce cell death of neighboring cells by the release of endolysin. The data presented are clear and convincing.

The novelty of the finding is a bit restricted as in many Gram-positive and -negative MV were described. One thinks only of the numerous publications on prophage-encoded endolysins causing hole formation and ghosts in *E. coli*. The authors themselves (Turnbull et al. 2016) have published recently already a related mechanism of endolysin-triggered MV formation accompanied by cell lysis in a stressed subpopulation of *Pseudomonas aeruginosa*. Also the Gram-positive *Staphylococcus aureus* produces MVs during both in vitro culture and in vivo infection that induce host cell death (Gurung, M. et al. 2011. *Staphylococcus aureus* produces membrane-derived vesicles that induce host cell death. PLoS One).

Nevertheless, the authors pinpoint the bubbling and MV formation to the SP β prophage encoded endolysin and holin genes, as responsible for the observed phenotype.

There are some questions that should be dealt with:

- 1) It would be interesting to know how many cells contain the pinholes; can they be quantified? pinholes/cell or (cells containing pinholes) or pinholes / 100 cells in WT, xhIAB-xlyA deletion and overexpression of xhIAB-xlyA.
- 2) The authors speak of cell lysis after induction with xylose, especially when long incubation was done, can this be quantified by CFU counts of induced vs. uninduced cells; how many cells (%) were lysed?

Minor points:

- 1) In Fig. 2E one cells gets stained by SYTOX but then seems to be destained again, without forming MVs, does that mean that the mechanism is reversible?
- 2) Do the authors have an explanation why only the combination of xhIAB-xlyA, but not xhIB-xlyA or xlyA increase MV formation? Do the proteins form a complex?
- 3) Extended Data Figure 1: In b) TEM of treated cells is shown. The cells, looks already completely lysed?! This is supposed to be 12.5% MMC, is the fold change of MV production in a) caused by total disruption of cells?
- 4) Line 118-120: This is difficult to understand, since the PBSX system is expressed in the deletion mutant? Please clarify what was done in this experiment and explain better in the text.
- 5) I am lacking an explanation why a *B. subtilis* cell should induce this bubbling cell death - can a possible explanation be discussed in the manuscript?

Reviewer #3 (Remarks to the Author):

In this manuscript the authors show that DNA damage stimulates MV formation in a RecA dependent manner involving a prophage derived endolysin. Conditional expression of the endolysin in exponentially growing cells stimulated MV formation. Expression of the endolysin is associated with generation of pinholes in the cell wall through which membrane material was extruded. They speculate that released endolysin promotes MV formation in neighboring cells.

Comments:

1. The authors demonstrate that in *B. subtilis*, exposure to a peptidoglycan damaging agent added exogenously like lysozyme or maybe cell released endolysins as well as endogenously produced endolysin, enhance MV formation. Basically, the manuscript reports increase MV formation upon

peptidoglycan damage. Which it has been reported as a way to increase MVs production in Streptococcus.

2. The title suggest a new insight on MV formation but the mechanism proposed is under a specific condition of stress which is interesting but the title should reflect that. Something like "Increased vesicle formation in the gram positive bacterium Bacillus subtilis, triggered by a prophage derived endolysin and peptidoglycan damage", is more accurate.

3. It is mentioned that without stress, cells that are producing large number of MVs die. What proportion of non-stressed cells produce MVs and what proportion of those die?

5. Is there endolysin included in the vesicles? If MV in the supernatant of induced cells are removed by ultracentrifugation does it loose the MV formation stimulating effect?

4. Is there an effect of the prophage deletions on MVs production without MMC stress?

5. lines 190-191; Change to: MV formation in the gram positive bacterium B. subtilis under DNA stress is dependent on the expression of a bacteriophage-derived endolysin.

Response to reviewers

> Reviewer #1 (Remarks to the Author):

> The paper by Toyofuku et al describes phage mediated lysis in *B. subtilis* and goes on to claim that this is a mechanism for extracellular vesicle formation. The paper does a disservice to the field by mixing up very different processes and bringing them together in a form that will sow confusion. Phage mediated lysis will result in damaged membranes, cell lysis and release of membrane fragments that can reassemble into structures that resemble vesicles as a function of the tendency of lipids to form bilayers and vesicles (like is seen in liposomes). However, to claim that this lysis is a mechanism for the phenomenon of extracellular MV vesicle formation that has now been described for many gram positive organisms is to jump to a conclusion that is not supported by the data. Hence, this paper should not be published in its current form. In essence there are serious concerns about the data shown in this paper and the extrapolations and conclusions made from those results to MV biogenesis are not supported by the data.

We agree that there are different routes for vesicle biogenesis and it is a matter of definition what a “true” vesicle rather than structures that resemble vesicles are. We feel that phage-triggered cell death is an underestimated mechanism for vesicle biogenesis in nature. This hypothesis is supported by recent studies that have demonstrated that vesicles can harbor complete viral genomes¹ and that DNA associated with MVs isolated from open ocean samples are highly enriched for viral sequences^{2,3}. Moreover, UV radiation of water samples, a trigger for prophage induction, was recently demonstrated to greatly stimulate vesicle formation⁴. Given that bacteriophages are the most abundant organisms in the biosphere and they are a ubiquitous feature of bacterial existence⁵, it is not unreasonable that naturally occurring vesicles (or structures that are described as vesicles in these publications) originate to a large extent from cell lysis. Moreover, MVs derived from chemically or mechanically lysed cells are already used as vaccines. We have added these informations to the Discussion.

However, the criticism of the reviewer was noticed and we now also mention other described mechanisms that can lead to vesicle formation.

> Specific comments

> 1. The title is misleading for it implies that MVs come from cell lysis. Many studies of gram positives have shown that MVs from gram positives are made by an active process from live cells. Phage- triggered lysis will produce vesicles from membrane association but the authors have produce no biochemical evidence that these vesicles are the same that are produced by non-lytic cells.

We agree with the reviewer and have changed the title to “Prophage-triggered vesicle formation through peptidoglycan damage in *Bacillus subtilis*” to indicate that this paper describes vesicles that originate from cell lysis. There are papers that describe

vesicle formation from supposedly live cells. However, these studies did not specifically investigate the possibility that the vesicles could originate from a subpopulation of lysed cells. A recent paper showed that vesicles produced by *Streptococcus pneumoniae* are specifically enriched for a putative phage-associated endolysin⁶. More important in the context of our study is the finding that a proteome analysis of vesicles produced by *B. subtilis* in standard laboratory conditions identified various phage proteins⁷. This enrichment was particularly high for sporulating *B. subtilis* cultures, which may be due to autolysin-triggered cell lysis. We have added this new information in the discussion.

> 2. Line 180 states 'In support of this hypothesis and in agreement with previous work^{23,24,26} we observed that cell wall weakening treatment with lysozyme stimulated MV formation; this is circular reasoning: lysozyme is known to lyse cells releasing membranes that can assemble into structures resembling vesicles. The same can be achieved by sonicating cells.

Cell wall weakening treatments have previously been used in several studies to induce vesicle formation in Gram-positive bacteria. Importantly, Biagini et al.⁸ showed that despite weakening the cell wall by treatment with a sublethal concentration of penicillin no significant changes in either the MV proteome or membrane lipidome could be observed.

> 3. Figure 1 purports to show vesicles. What is the evidence that the structures pointed to by the arrows are vesicles? I can easily argue that these are sectional fractions of cells that appear smaller by microscopy. Figure 1 data is not convincing. This reviewer has the same concern for Figure 2e – I can argue that what appears to be vesicle is nothing more than a cross-section of a bacterium that was in the background and has moved into view. The microscopy shown is substandard for the claims being made.

The fluorescence microscopy was performed with cells grown on agarose pads, where they formed fine monolayers. We have uploaded a 3D reconstruction of the confocal microscopy image in figure 1 (Fig.1_3D_For_reviewing_purpose_only.mov) that clearly shows that the arrows do not point at a cross section of a bacterium. However, the criticism was noticed and we have also improved our microscopy for further evidence. Using cryo-electron tomography we were for the first time able to resolve vesicle formation in three dimensions at the nanometer scale. The 3D reconstruction clearly shows how inner membrane material protrudes through holes in the cell wall in a strain expressing the phage endolysin. We have added this additional analysis along with the Material and Methods in the revised version of the manuscript.

> 4. Extended figure 3d purports to show MVs by EM. These images do not look like the spherical vesicle populations described in other papers of gram+ MV. To this author these structures look like lysed membrane fragments, which are very different than MVs.

As stated in the Material and Methods, the MVs were isolated and purified according to well established protocols. In fact, our TEM images of *B. subtilis* MVs are very similar to what have been observed in previous studies^{9,10}. SEM images, which have been added as Supplementary Fig. 4c to the revised manuscript, also show the typical

blebbing of the cell surface that has been observed earlier for MV-producing *B. subtilis* cells⁹. To visualize the fine structure of these vesicles we have also employed cryo-electron tomography and these new images have been included in the revised manuscript.

> 5. The term 'bubbling cell death' makes no sense. What the authors are describing is bacterial cell lysis from phage and/or endogenous lysins...there is no need for additional confounding terminology.

We have removed this term.

> Reviewer #2 (Remarks to the Author):

>

> The authors describe an interesting mechanism for MV formation in *B. subtilis*. They show the formation of so called pinholes in the PGN of prophage PBSX containing cells of *B. subtilis*. The prophage encoded endolysin gene is SOS-inducible and responsible for pinhole formation that causes finally cell death due to cytoplasmic membrane damage. The formed MV can induce cell death of neighboring cells by the release of endolysin. The data presented are clear and convincing.

> The novelty of the finding is a bit restricted as in many Gram- positive and –negative MV were described. One thinks only of the numerous publications on prophage-encoded endolysins causing hole formation and ghosts in *E. coli*. The authors themselves (Turnbull et al. 2016) have published recently already a related mechanism of endolysin- triggered MV formation accompanied by cell lysis in a stressed subpopulation of *Pseudomonas aeruginosa*. Also the Gram-positive *Staphylococcus aureus* produces MVs during both in vitro culture and in vivo infection that induce host cell death (Gurung, M. et al. 2011. *Staphylococcus aureus* produces membrane-derived vesicles that induce host cell death. PLoS One).

>

Our recently proposed endolysin-based mechanism for vesicle biogenesis of Gram-negative bacteria has been enthusiastically accepted and promoted by many researches, as seen by the large number of mentions in news and blogs and an excellent altmetric score of 154. However, we also realized that some researchers are unhappy with our model, as it does not promote the commonly accepted view that vesicle formation follows a genetically hard-wired program and represents a specific mechanism for the export of virulence factors and other public goods. In fact, we are not opposing such a model, but our results and the data of other recent publications just provide strong evidence that at least a large proportion of the vesicles present in natural habitats originate from cell lysis (see above).

Given this current controversy, this paper would add a lot of weight to the lysis-based mechanism of vesicle biogenesis that may help to convince the scientific community. It is also important to note that to our knowledge, no confirmed mechanism for MV biogenesis in Gram-positive cells exists. In a Nature Microbiology review from 2015 the authors present three hypotheses to explain how MVs traverse thick cell walls but conclude that “current knowledge in the field is mainly limited to basic studies that characterize extracellular vesicles but lack mechanistic insight”¹¹.

> Nevertheless, the authors pinpoint the bubbling and MV formation to the SP β prophage encoded endolysin and holin genes, as responsible for the observed phenotype.

> There are some questions that should be dealt with:

> 1) It would be interesting to know how many cells contain the pinholes; can they be quantified? pinholes/cell or (cells containing pinholes) or pinholes / 100 cells in WT, xhIAB-xlyA deletion and overexpression of xhIAB-xlyA.

It is possible to count the number of cells containing pinholes in the TEM images shown in Fig 3b. However, given that this image is a thin section of a cell, we will

severely underestimate the number of cells containing pinholes. From the analysis of the holin-endolysin promoter activity shown in Fig. 2c and Supplementary Fig. 2 we estimate that approximately 2.5% of the cells of a wild type culture contain pinholes under non-stress conditions.

> 2) The authors speak of cell lysis after induction with xylose, especially when long incubation was done, can this be quantified by CFU counts of induced vs. uninduced cells; how many cells (%) were lysed?

A growth curve and CFU counts were added as Supplementary Fig 3 c and d, showing that the number of cells lysing depend on time of induction and the concentration of xylose.

> Minor points:

>1) In Fig. 2E one cells gets stained by SYTOX but then seems to be destained again, without forming MVs, does that mean that the mechanism is reversible?

Most likely we were just unable to capture the release of vesicles from this cell. The destaining is indicative of the release of DNA from a dead cell. DNA release is more clearly shown in Fig 1 and the Supplementary fig. 5, supporting this idea.

> 2) Do the authors have an explanation why only the combination of xhIAB-xlyA, but not xhIB-xlyA or xlyA increase MV formation? Do the proteins form a complex?

Previous work has shown that cell only lyse when all three proteins are expressed¹². The function of XhIA is unknown but is predicted to be a membrane protein that forms a complex with XhIB. We have added this information to the manuscript.

> 3) Extended Data Figure 1: In b) TEM of treated cells is shown. The cells, looks already completely lysed?! This is supposed to be 12.5% MMC, is the fold change of MV production in a) caused by total disruption of cells?

We are sorry for the confusion. In this experiment, MVs were separated from the cells and purified from the supernatant. Hence the membranous spherical structures observed in this figure are MVs but not cells. We have added the term “MV” in the figure legend.

> 4) Line 118-120: This is difficult to understand, since the PBSX system is expressed in the deletion mutant? Please clarify what was done in this experiment and explain better in the text.

We are sorry that this section was confusing. To improve clarity we have rephrased this part of the Results section as follows: We used a transcriptional fusion of the PBSX holin-endolysin promoter (P_L promoter) to the reporter gene *zsGreen* to

determine the fraction of cells that entered the lytic cycle. This analysis revealed that 5.22 ± 0.50 % of the cells of the $\Delta xhIAB-xlyA$ mutant and 2.45 ± 0.32 % of the wild-type cells were induced in liquid LB medium. The reduced number of fluorescent cells in the wild type background is indicative of lysis due to the expression of the holin-endolysin system and the concomitant release of the fluorescent reporter protein

> 5) I am lacking an explanation why a *B. subtilis* cell should induce this bubbling cell death - can a possible explanation be discussed in the manuscript?

Previous work has shown that MV production is an important factor in neutralizing environmental agents that target the outer membrane of Gram-negative bacteria, such as antimicrobial peptides or bacteriophages¹³. Furthermore, cell lysis leads to DNA release that promotes the formation of biofilms, where bacteria show greatly increased resistance to various stresses¹⁴. Hence, although endolysin-triggered vesicle formation is linked to the death of some cells this can provide a benefit to the rest of the population. We have added this information to the discussion.

> Reviewer #3 (Remarks to the Author):

> In this manuscript the authors show that DNA damage stimulates MV formation in a RecA dependent manner involving a prophage derived endolysin. Conditional expression of the endolysin in exponentially growing cells stimulated MV formation. Expression of the endolysin is associated with generation of pinholes in the cell wall through which membrane material was extruded. They speculate that released endolysin promotes MV formation in neighbouring cells.

> Comments:

>1. The authors demonstrate that in *B. subtilis*, exposure to a peptidoglycan damaging agent added exogenously like lysozyme or maybe cell released endolysins as well as endogenously produced endolysin, enhance MV formation. Basically, the manuscript reports increase MV formation upon peptidoglycan damage. Which it has been reported as a way to increase MVs production in *Streptococcus*.

Exposure to PG damaging agents, including chemical or mechanical treatment, is routinely used to generate bacterial MVs, particularly for the mass production of MV vaccines. However, the situation described in our manuscript is different as the PG damage is induced by the cell itself or by its neighbouring cells and possibly reflects the natural situation in the environment.

> 2. The title suggest a new insight on MV formation but the mechanism proposed is under a specific condition of stress which is interesting but the title should reflect that. Something like "Increased vesicle formation in the gram positive bacterium *Bacillus subtilis*, triggered by a prophage derived endolysin and peptidoglycan damage", is more accurate.

We agree that this would be a very appropriate title. However, according to Nature Communications guidelines the title should be 15 words or fewer. As a possible title we would therefore suggest: "Prophage-triggered membrane vesicle formation through peptidoglycan damage in *Bacillus subtilis*"

> 3. It is mentioned that without stress, cells that are producing large number of MVs die. What proportion of non-stressed cells produce MVs and what proportion of those die?

We have quantified the proportion of cells that die without any obvious stress and this information has been added in the text.

> 5. Is there endolysin included in the vesicles? If MV in the supernatant of induced cells are removed by ultracentrifugation does it loose the MV formation stimulating effect?

We have added data (Fig. 4d) showing that MVs carry endolysin can induce MV formation in other cells. The MV-free supernatant also stimulated MV formation, indicating that not all of the endolysin is packaged into MVs.

> 4. Is there an effect of the prophage deletions on MVs production without MMC stress?

Similar to *P. aeruginosa*¹⁴, we do not see a difference between the wildtype and the PBSX mutant when grown in liquid medium. However, the importance of the holin-endolysin system in MV production becomes evident when cells are under conditions where the SOS response is activated. The data showing MV formation under no-stress condition is added as Supplementary Figure 1f.

> 5. lines 190-191; Change to: MV formation in the gram positive bacterium *B. subtilis* under DNA stress is dependent on the expression of a bacteriophage-derived endolysin.

We have added the term under DNA stress.

References

- 1 Gaudin, M. *et al.* Extracellular membrane vesicles harbouring viral genomes. *Environ Microbiol* **16**, 1167-1175, doi:10.1111/1462-2920.12235 (2014).
- 2 Biller, S. J. *et al.* Bacterial vesicles in marine ecosystems. *Science* **343**, 183-186, doi:10.1126/science.1243457 (2014).
- 3 Soler, N., Krupovic, M., Marguet, E. & Forterre, P. Membrane vesicles in natural environments: a major challenge in viral ecology. *ISME J.*, doi:10.1038/ismej.2014.184 (2014).
- 4 Gamalier, J. P., Silva, T. P., Zarantonello, V., Dias, F. F. & Melo, R. C. Increased production of outer membrane vesicles by cultured freshwater bacteria in response to ultraviolet radiation. *Microbiol Res.* **194**, 38-46, doi:10.1016/j.micres.2016.08.002 (2017).
- 5 Clokie, M. R., Millard, A. D., Letarov, A. V. & Heaphy, S. Phages in nature. *Bacteriophage* **1**, 31-45, doi:10.4161/bact.1.1.14942 (2011).
- 6 Resch, U. *et al.* A Two-Component Regulatory System Impacts Extracellular Membrane-Derived Vesicle Production in Group A Streptococcus. *MBio* **7**, doi:10.1128/mBio.00207-16 (2016).
- 7 Kim, Y., Edwards, N. & Fenselau, C. Extracellular vesicle proteomes reflect developmental phases of *Bacillus subtilis*. *Clin. Proteomics* **13**, 6, doi:10.1186/s12014-016-9107-z (2016).
- 8 Biagini, M. *et al.* The Human Pathogen *Streptococcus pyogenes* Releases Lipoproteins as Lipoprotein-rich Membrane Vesicles. *Mol. Cell. Proteomics* **14**, 2138-2149, doi:10.1074/mcp.M114.045880 (2015).
- 9 Brown, L., Kessler, A., Cabezas-Sanchez, P., Luque-Garcia, J. L. & Casadevall, A. Extracellular vesicles produced by the Gram-positive bacterium *Bacillus subtilis* are disrupted by the lipopeptide surfactin. *Mol. Microbiol.* **93**, 183-198, doi:10.1111/mmi.12650 (2014).
- 10 Tzipilevich, E., Habusha, M. & Ben-Yehuda, S. Acquisition of Phage Sensitivity by Bacteria through Exchange of Phage Receptors. *Cell* **168**, 186-199 e112, doi:10.1016/j.cell.2016.12.003 (2017).
- 11 Brown, L., Wolf, J. M., Prados-Rosales, R. & Casadevall, A. Through the wall: extracellular vesicles in Gram-positive bacteria, mycobacteria and fungi. *Nat. Rev. Microbiol.* **13**, 620-630, doi:10.1038/nrmicro3480 (2015).
- 12 Krogh, S., Jorgensen, S. T. & Devine, K. M. Lysis genes of the *Bacillus subtilis* defective prophage PBSX. *J. Bacteriol.* **180**, 2110-2117 (1998).
- 13 Manning, A. J. & Kuehn, M. J. Contribution of bacterial outer membrane vesicles to innate bacterial defense. *BMC Microbiol.* **11**, 258, doi:10.1186/1471-2180-11-258 (2011).
- 14 Turnbull, L. *et al.* Explosive cell lysis as a mechanism for the biogenesis of bacterial membrane vesicles and biofilms. *Nat. Commun.* **7**, 11220, doi:10.1038/ncomms11220 (2016).

REVIEWERS' COMMENTS:

Reviewer #1 (Remarks to the Author):

The authors have done a good job in revising this manuscript. The revised title and wording now makes it clear that vesicle formation from phage-induced lysis is one mechanism for extracellular vesicle formation. It is unlikely that this mechanism applies to most of the reports of vesicle formation in gram positive bacteria in the literature given that lipid analysis of extracellular vesicles has shown major differences from bacterial membranes and that for several bacteria their toxins have been shown to be packaged in vesicles. Furthermore, vesicles from bacterial lysis and those recovered in growing culture differ in heterogeneity and size. In the opinion of this reviewer the authors are still pushing a bit hard on the phage lysis explanation as a general mechanism for extracellular vesicle production. For example, if you read the abstract one can come away with the impression that they have found a general system for vesicle formation and yet have studied only one system and carried out no biophysical or biochemical analysis of the vesicles. I think more caution in claims by tweaking several sentences would make this a better paper and avoid fueling the kinds of controversies that have plagued the vesicle field. My recommendation would be to do that.

Reviewer #2 (Remarks to the Author):

[No further comments for author.]

Reviewer #3 (Remarks to the Author):

The additional tomography helps to make the point that PG damage leads to membrane material extruded through the cell wall, rounded up and eventually detached from the cell wall as a membrane vesicle.

Please remove "it is likely" from line 239. There is sufficient evidence that non-dying bacteria release MVs. The authors own data indicate that even the recA mutant has reduced but not eliminated MVs release. Presumably, those are endolysin independent.

The manuscript emphasizes the importance of endolysins in MVs formation.

It would be interesting to discuss any ideas the authors might have on whether a controlled, perhaps localized, non lethal PG alteration could aid MVs release from thick cell wall bacteria that are not destined to die.

Reviewer #1 (Remarks to the Author):

The authors have done a good job in revising this manuscript. The revised title and wording now makes it clear that vesicle formation from phage-induced lysis is one mechanism for extracellular vesicle formation. It is unlikely that this mechanism applies to most of the reports of vesicle formation in gram positive bacteria in the literature given that lipid analysis of extracellular vesicles has shown major differences from bacterial membranes and that for several bacteria their toxins have been shown to be packaged in vesicles. Furthermore, vesicles from bacterial lysis and those recovered in growing culture differ in heterogeneity and size. In the opinion of this reviewer the authors are still pushing a bit hard on the phage lysis explanation as a general mechanism for extracellular vesicle production. For example, if you read the abstract one can come away with the impression that they have found a general system for vesicle formation and yet have studied only one system and carried out no biophysical or biochemical analysis of the vesicles. I think more caution in claims by tweaking several sentences would make this a better paper and avoid fueling the kinds of controversies that have plagued the vesicle field. My recommendation would be to do that.

>We have further tuned down our statement and have modified the manuscript according to the editor's suggestions to avoid the impression that phage-induced lysis is the only mechanism for MV formation in bacteria.

We agree with the reviewer that MVs are heterogeneous in composition, size and also how they are formed. The heterogeneity of MVs may also have led to the controversy in the field of how they are formed, as different mechanisms may lead to different MV types. Ongoing work aims at identifying structural and compositional differences of MVs depending on their origin. With this knowledge it will also be possible to estimate the contribution of the different pathways for vesicle formation in natural and clinical environments. However, this is future work and will be a story on its own.

Reviewer #2 (Remarks to the Author):

[No further comments for author.]

Reviewer #3 (Remarks to the Author):

The additional tomography helps to make the point that PG damage leads to membrane material extruded through the cell wall, rounded up and eventually detached from the cell wall as a membrane vesicle.

Please remove "it is likely" from line 239. There is sufficient evidence that non-dying bacteria release MVs. The authors own data indicate that even the recA mutant has reduced but not eliminated MVs release. Presumably, those are endolysin independent.

The manuscript emphasizes the importance of endolysins in MVs formation. It would be interesting to discuss any ideas the authors might have on whether a controlled, perhaps localized, non lethal PG alteration could aid MVs release from thick cell wall bacteria that are not destined to die.

> We have removed the phrase "it is likely".

While the idea that a controlled PG alteration could aid MV release in Gram-positive bacteria without lysing the cells is attractive, there is currently no evidence that such a mechanism exists. We therefore feel reluctant to speculate about such possible mechanisms. We agree with the reviewer that other mechanisms of MV biogenesis exist but they are currently unknown.